# Reconstruct & Crush Network

**Erinç Merdivan**[1,2]**, Mohammad Reza Loghmani**[3] **and Matthieu Geist**[4]

[1] AIT Austrian Institute of Technology GmbH, Vienna, Austria
[2] LORIA (Univ. Lorraine & CNRS), CentraleSupélec, Univ. Paris-Saclay, 57070 Metz, France
[3] Vision4Robotics lab, ACIN, TU Wien, Vienna, Austria
[4] Université de Lorraine & CNRS, LIEC, UMR 7360, Metz, F-57070 France
`erinc.merdivan@ait.ac.at, loghmani@acin.tuwien.ac.at`
`matthieu.geist@univ-lorraine.fr`

## Abstract

This article introduces an energy-based model that is adversarial regarding data: it minimizes the energy for a given data distribution (the positive samples) while maximizing the energy for another given data distribution (the negative or unlabeled samples). The model is especially instantiated with autoencoders where the energy, represented by the reconstruction error, provides a general distance measure for unknown data. The resulting neural network thus learns to reconstruct data from the first distribution while crushing data from the second distribution. This solution can handle different problems such as Positive and Unlabeled (PU) learning or covariate shift, especially with imbalanced data. Using autoencoders allows handling a large variety of data, such as images, text or even dialogues. Our experiments show the flexibility of the proposed approach in dealing with different types of data in different settings: images with CIFAR-10 and CIFAR-100 (not-in-training setting), text with Amazon reviews (PU learning) and dialogues with Facebook bAbI (next response classification and dialogue completion).

## 1 Introduction

The main purpose of machine learning is to make inferences about unknown data based on encoded dependencies between variables learned from known data. Energy-based learning [16] is a framework that achieves this goal by using an energy function that maps each point of an input space to a single scalar, called *energy*. The fact that energy-based models are not subject to the normalizability condition of probabilistic models makes them a flexible framework for dealing with tasks such as prediction or classification.

In the recent years, with the advancement of deep learning, astonishing results have been achieved in classification [15, 25, 8, 26]. These solutions focus on the standard setting, in which the classifier learns to discriminate between $K$ classes, based on the underlying assumption that the training and test samples belong to the same distribution. This assumption is violated in many applications in which the dynamic nature [6] or the high cardinality [19] of the problem prevent the collection of a representative training set. In the literature, this problem is referred to as *covariate shift* [7, 24].

In this work, we address the covariate shift problem by explicitly learning features that define the intrinsic characteristics of a given class of data rather than features that discriminate between different classes. The aim is to distinguish between samples of a positive class ($A$) and samples that do not belong to this class ($\neg A$), even when test samples are not drawn from the same distribution as the training samples. We achieve this goal by introducing an energy-based model that is adversarial regarding data: it minimizes the energy for a given data distribution (the positive samples) while maximizing the energy for another given data distribution (the negative or unlabeled samples). The model is instantiated with autoencoders because of their ability to learn data manifolds.

In summary, our contributions are the following:

- a simple energy-based model dealing with the $A/\neg A$ classification problem by providing a distance measure of unknown data as the energy value;
- a general framework that can deal with a large variety of data (images, text and sequential data) by using features extracted from an autoencoder architecture;
- a model that implicitly addresses the imbalanced classification problem;
- state-of-the-art results for the dialogue completion task on the Facebook bAbI dataset and competitive results for the general $A/\neg A$ classification problem using different datasets such as CIFAR-10, CIFAR-100 and Amazon Reviews.

The next section introduces the proposed "reconstruct & crush" network, section 3 positions our approach compared to related works, section 4 presents the experimental results on the aforementioned problems and section 5 draws the conclusions.

## 2   Model

Let define $p_{\text{pos}}$ as the probability distribution producing positive samples, $x_{\text{pos}} \sim p_{\text{pos}}$. Similarly, write $p_{\text{neg}}$ the distribution of negative samples, $x_{\text{neg}} \sim p_{\text{neg}}$. More generally, these negative samples can be *unlabeled* samples (possibly containing positive samples). This case will be considered empirically, but we keep this notation for now.

Let $N$ denote a neural network that takes as input a sample $x$ and outputs a (positive) energy value $E$:
$$N(x) = E \in \mathbb{R}^+.$$

The proposed approach aims at learning a network $N$ that assign low energy values to positive samples ($N(x_{\text{pos}})$ small for $x_{\text{pos}} \sim p_{\text{pos}}$) and high energy values for negative samples ($N(x_{\text{neg}})$ high for $x_{\text{neg}} \sim p_{\text{neg}}$).

Let $m > 0$ be a user-defined margin, we propose to use the following loss $\mathcal{L}_N$ and associated risk $\mathcal{R}_N$:

$$\mathcal{L}(x_{\text{pos}}, x_{\text{neg}}; N) = N(x_{\text{pos}}) + \max(0, m - N(x_{\text{neg}}))$$
$$\mathcal{R}(N) = \mathbb{E}_{x_{\text{pos}} \sim p_{\text{pos}}, x_{\text{neg}} \sim p_{\text{neg}}} \mathcal{L}(x_{\text{pos}}, x_{\text{neg}})$$
$$= \mathbb{E}_{x_{\text{pos}} \sim p_{\text{pos}}}[N(x_{\text{pos}})] + \mathbb{E}_{x_{\text{neg}} \sim p_{\text{neg}}}[\max(0, m - N(x_{\text{neg}}))]. \qquad (1)$$

Ideally, minimizing this risk amounts to have no reconstruction error over positive samples and a reconstruction error greater than $m$ (in expectation) over negative samples. The second term of the risk acts as a regularizer that enforces the network to assign a low energy only to positive samples. The choice of the margin $m$ will affect the behavior of the network: if $m$ is too small a low energy will be assigned to all inputs (both positive and negative), while if $m$ is too large assigning a large energy to negative samples can prevent from reconstructing the positive ones.

We specialize our model with autoencoders, that are a natural choice to represent energy-based models. An autoencoder is composed of two parts, the encoder ($\mathrm{Enc}$) that projects the data into an encoding space, and the decoder ($\mathrm{Dec}$) that reconstructs the data from this projection:

$$\mathrm{Enc} : \mathcal{X} \rightarrow \mathcal{Z}$$
$$\mathrm{Dec} : \mathcal{Z} \rightarrow \mathcal{X}$$
$$\underset{\mathrm{Enc}, \mathrm{Dec}}{\operatorname{argmin}} \|x - \mathrm{Dec}(\mathrm{Enc}(x))\|^2.$$

Here, $\mathcal{X}$ is the space of the input data (either positive or negative) and $\mathcal{Z}$ is the space of encoded data. In this setting, the reconstruction error of a sample $x$ can be interpreted as the energy value associated to that sample:

$$N(x) = \|x - \mathrm{Dec}(\mathrm{Enc}(x))\|^2 = E.$$

Our resulting reconstruct & crush network (RCN) is thus trained to assign a low reconstruction error to $x_{\text{pos}}$ (*reconstruct*) and an high reconstruction error to $x_{\text{neg}}$ (*crush*).

Any stochastic gradient descent method can be used to optimize the risk of Eq. (1), the mini-batches of positive and negative samples being sampled independently from the corresponding distributions.

# 3 Related work

With the diffusion of deep neural networks, autoencoders have received a new wave of attention due to their use for layer-wise pretraining [1]. Although the concept of autoencoders goes back to the 80s [23, 3, 10], many variations have been proposed more recently, such as denoising autoencoder [27], stacked autoencoders [9] or variational autoencoders [13].

Despite the use of autoencoders for pretraining is not a common practice anymore, various researches still take advantage of their properties. In energy-based generative adversarial networks (EBGAN) [30], an autoencoder architecture is used to discriminate between real samples and "fake" ones produced by the generator. Despite not being a generative model, our method shares with EBGAN the interpretation of the reconstruction error provided by the autoencoder as energy value and the fundamentals of the discriminator loss. However, instead of the samples produced by the generator network, we use negative or unlabeled samples to push the autoencoder to discover the data manifold during training. In other words, EBGAN searches for a generative model by training adversarial networks, while in our framework the network tries to make two distributions adversarial.

The use of unlabeled data (that could contain both positive and negative samples) together with positive samples during training is referred to as PU (Positive and Unlabeled) learning [5, 17]. In the literature, works in the PU learning setting [29, 18] focus on text-based applications. Instead, we show in the experiments that our work can be applied to different type of data such as images, text and sequential data.

Similarly to our work, [11] uses the reconstruction error as a measure to differentiate between positive and negative samples. However they train their network with either positive or negative data only. In addition, instead of end-to-end training, they provide a two-stage process in which a classifier is trained to discriminate between positive and negative samples based on the reconstruction error.

In the context of dialogue management systems, the score proposed in [21] has been used as a quality measure of the response. Nevertheless, [19] shows that this score fails when a correct response, that largely diverges from the ground truth, is given. The energy value of the RCN is a valid score to discriminate between good and bad responses, as we show in section 4.4.

# 4 Experimental results

In this section, we experiment the proposed RCN on various tasks with various kind of data. We consider a not-in-training setting for CIFAR-10 and CIFAR-100 (sections 4.1 and 4.2), a PU learning setting for the amazon reviews dataset (section 4.3) and a dialogue completion setting for the Facebook bAbI dataset (section 4.4).

For an illustrative purpose, we also provide examples of reconstructed and crushed images from CIFAR-10 and CIFAR-100 in figure 1, corresponding to experiments of sections 4.1 and 4.2.

## 4.1 CIFAR-10

CIFAR-10 consists of 60k 32x32 color images in 10 classes, with 6k images per class. There are 50k training images and 10k test images [14]. We converted the images to gray-scale and used 5k images per class.

This set of experiments belong to the not-in-training setting [6]: the training set contains positive and negative samples and the test set belongs to a different distribution than the training set. The "automobile" class is used as the positive class ($A$) and the rest of the classes are considered to be the negative class ($\neg A$) (binary classification problem). All the training samples are used for training, except for those belonging to the "ship" class. Test samples of "automobile" and "ship" are used for testing. It is worth noticing that the size of positive and negative training sets is highly imbalanced: 5k positive samples and 40k negative samples.

In this experiment, we show the superior performances of our network with respect to standard classifiers in dealing with images of an unseen class. Since we are dealing with a binary classification problem, we define a threshold $T$ for the energy value. This threshold is used in RCN to distinguish between the positive and the negative class. For our autoencoder, we used a convolutional network defined as: (32)3c1s-(32)3c1s-(64)3c2s-(64)3c2-(32)3c1s-512f-1024f, where "(32)3c1s" denotes

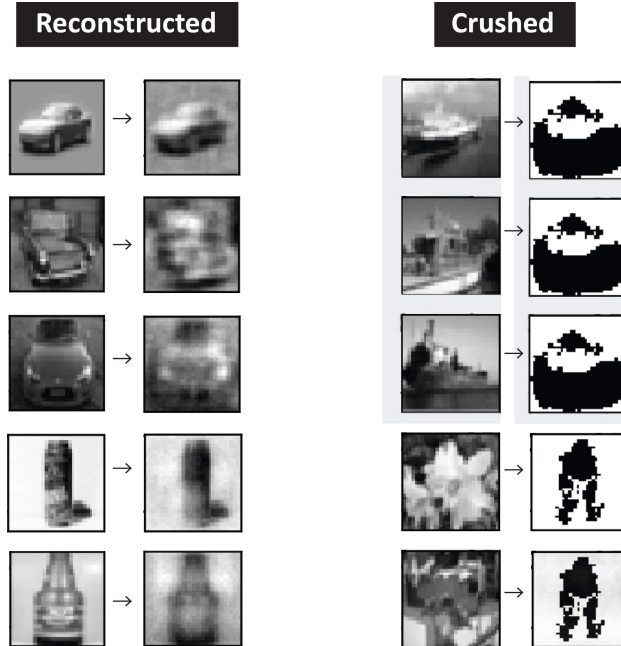

Figure 1: Illustrations of Reconstructed and Crushed images by RCN from CIFAR10 and CIFAR100.

a convolution layer with 32 output feature maps, kernel size 3 and stride 1, and "512f" denotes a fully-connected layer with 512 hidden units. The size of the last layer corresponds to the size of the images (32x32=1024). For standard classification we add on top of the last layer another fully-connected layer with 2 output neurons ($A/\neg A$). The choice of the architectures for standard classifier and autoencoder is driven by necessity of fair comparison. ReLU activation functions are used for all the layers except for the last fully-connected layer of the standard classifier in which a Softmax function is used. These models are implemented in Tensorflow and trained with the adam optimizer [12] (learning rate of $0.0004$) and a mini-batch size of 100 samples. The margin $m$ was set to 1.0 and the threshold $T$ to 0.5.

Table 1 shows the true positive rate (TPR=#(correctly classified cars)/#cars) and the true negative rate (TNR=#(correctly classified ships)/#ships) obtained by the standard classifier (CNN / CNN-reduced) and our network (RCN). CNN-reduced shows the performance of the standard classifier when using the same amount of positive and negative samples. It can be noticed that RCN presents the best TNR and a TPR comparable to the one of CNN-reduced. These results shows that RCN is a better solution when dealing with not-in-training data. In addition, the TPR and TNR of our method is comparable despite the imbalanced training set.

Figure 2 clearly shows that not-in-training samples (ship images) are positioned between positive in-training samples (automobile images) and in-training-negative samples (images from all classes except automobile and ship). It can be noticed that negative in-training samples have a reconstruction loss close to margin value 1.0.

Table 1: Performances of standard classifier (CNN / CNN-reduced) and our method (RCN) on CIFAR-10. The positive class corresponds to "automobile" and the negative class corresponds to "ship" (unseen during the training phase).

| Method | True Positive Rate | True Negative Rate |
|---|---|---|
| CNN-reduced | 0.82 | 0.638 |
| CNN | 0.74 | 0.755 |
| RCN | 0.81 | 0.793 |

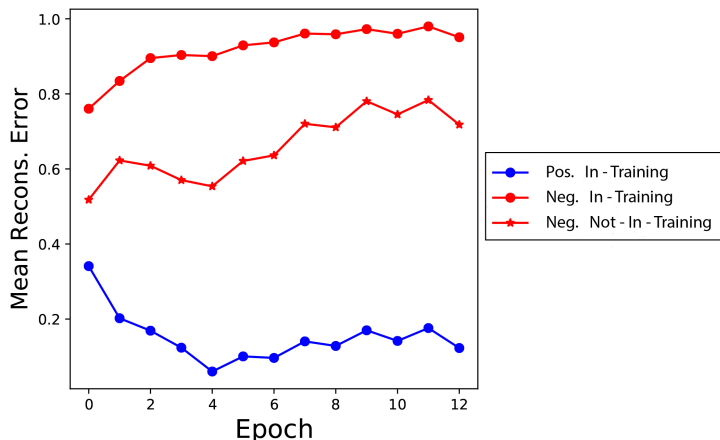

Figure 2: Mean reconstruction error over the epochs of positive in-training, negative in-training and negative not-in-training samples of CIFAR-10.

## 4.2 CIFAR-100

CIFAR-100 is similar to CIFAR-10, except it has 100 classes containing 600 images each (500 for training and 100 for testing) [14]. The 100 classes in the CIFAR-100 are grouped into 20 super-classes with 5 classes each. Each image comes with a pair of labels: the class and the super-class.

In this set of experiments, the "food containers" super-class is used as the positive class ($A$) and the all the other super-classes are considered to be the negative class ($\neg A$) (binary classification problem). During training, 4 out of 5 classes belonging to the "food containers" super-class ("bottles", "bowls", "cans", "cups") are used as the positive training set and 4 out of 5 classes belonging to the "flowers" super-class ("orchids", "poppies", "roses", "sunflowers") are used as the negative training set. At test time, two in-training classes ("cups" and "sunflowers"), two not-in-training classes belonging to "food containers" ("plates") and "flowers" ("tulips") and two not-in-training classes belonging to external super-classes ("keyboard" and "chair") are used.

In this experiment, we show the superior performances of our network with respect to standard classifiers in dealing with data coming from unknown distributions and from unseen modes of the same distributions as the training data. The same networks and parameters of section 4.1 are used here.

Table 2 shows the true positive rate (TPR=#(correctly classified plates)/#plates) and the true negative rate (TNR=#(correctly classified tulips)/#tulips) obtained by the standard classifier (CNN) and our network (RCN). It can be noticed that RCN presents the best results both for TNR and for TPR. These results shows that RCN is a better solution when dealing with not-in-training data coming from unseen modes of the data distribution. It is worth noticing that only 4k samples (2k positive and 2k negative) have been used during training.

Figure 3 clearly shows the effectiveness of the learning procedure of our framework: the networks assigns low energy value (close to 0) to positive samples, high energy value (close to $m$) to negative samples related to the negative training set and medium energy value (close to $m/2$) to negative samples unrelated to the negative training set.

Table 2: Performances of the standard classifier (CNN) and our method (RCN) on CIFAR-100. The positive class corresponds to "plates" and the negative class corresponds to "tulips".

| Method | True Positive Rate | True Negative Rate |
|--------|--------------------|--------------------|
| CNN    | 0.718              | 0.81               |
| RCN    | 0.861              | 0.853              |

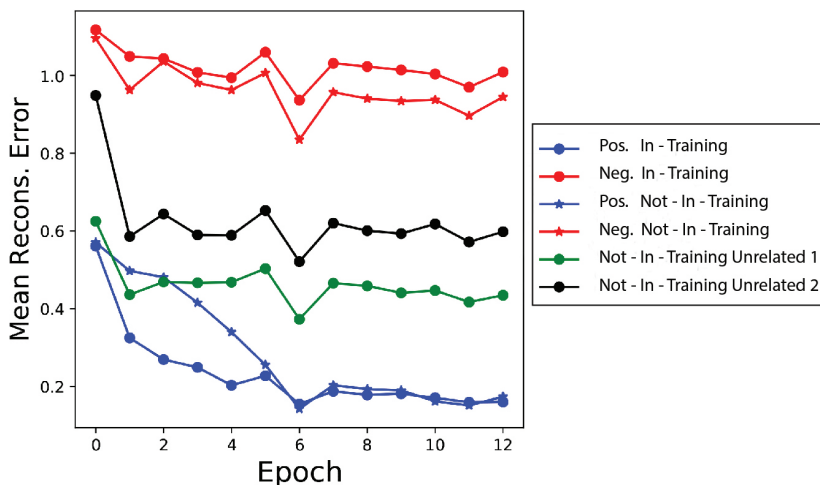

Figure 3: Mean reconstruction error over the epochs of positive in-training and not-in-training (blue), negative in-training and not-in-training (red) and not-in-training unrelated (green,black) of CIFAR-100.

## 4.3 Amazon review

Amazon reviews is a dataset containing product reviews (ratings, text, helpfulness votes) and meta-data (descriptions, category information, price, brand, and image features) from Amazon, including 142.8 million reviews spanning [20]. Here, we only use the ratings and text features.

This set of experiments belong to the PU learning setting: the training set contains positive and unlabeled data. The positive training set contains 10k "5-star" reviews and the unlabeled training set contains 10k unlabeled review (containing both positive and negative review). The test set is composed of 10k samples: 5k "5-star" (positive) reviews and 5k "1-star" (negative) reviews. The aim here is to show that RCN performs well in the PU learning setting with unlabeled sets with different positive/negative samples ratio.

For handling the text data, we used the pretrained Glove word-embedding [22] with 100 feature dimensions. We set the maximum number of words in a sentence to 40 and zero-padded shorter sentences.

For our autoencoder, we used a 1-dimensional (1D) convolutional network defined as: (128)7c1s-(128)7c1s-(128)3c1s-(128)3c1-(128)3c1s-2048f-4000f, where "(128)7c1s" denotes a 1D convolution layer with 128 output feature maps, kernel size 7 and stride 1. ReLU activation functions are used for all the layers. These models are implemented in Tensorflow and trained with the adam optimizer (learning rate of $0.0004$) and a mini-batch size of 100 samples. The margin $m$ was set to $0.85$ and the threshold $T$ to $0.425$.

Table 3 shows the results of different well-established PU learning methods, together with ours (RCN), on the Amazon review dataset. In can be noticed that, despite the fact that the architecture of our method is not specifically designed for handling the PU learning setting, it shows comparable results to the other methods, even when unlabeled training data with a considerable amount of positive samples ($50\%$) are used.

Table 4 presents some examples from the test set. It can be noticed that positive comments are assigned a low reconstruction error (energy) and vice-versa.

## 4.4 Facebook bAbI dialogue

Facebook bAbI dialogue is a dataset containing dialogues related to 6 different tasks in which the user books a table in a restaurant with the help of a bot [2]. For each task 1k training and 1k test dialogues are provided. Each dialogue has 4 to 11 turns between the user and the bot for a total of

Table 3: F-measure of positive samples obtained with Roc-SVM [28], Roc-EM [18], Spy-SVM [18], NB-SVM [18], NB-EM [18] and RCN (ours). The scores are obtained on two different configuration of the unlabeled training set: one containing 5% of positive samples and one containing 50% of positive samples.

| Method | F-measure for pos. samples (%5-%95) | F-measure for pos. samples (%50-%50) |
|---|---|---|
| Roc-SVM [28] | 0.92 | 0.89 |
| Roc-EM [18] | 0.91 | 0.90 |
| Spy-SVM [18] | 0.92 | 0.89 |
| NB-SVM [18] | 0.92 | 0.86 |
| NB-EM [18] | 0.91 | 0.86 |
| RCN | 0.90 | 0.83 |

Table 4: Examples of positive (5/5 score) and negative (1/5 score) reviews from Amazon review with the corresponding reconstruction error assigned from RCN.

| Review | Score | Error |
|---|---|---|
| excellent funny fast reading i would recommend to all my friends | 5/5 | 0.00054 |
| this is easily one of my favorite books in the series i highly recommend it | 5/5 | 0.00055 |
| super book liked the sequence and am looking forward to a sequel keeping the s and characters would be nice | 5/5 | 0.00060 |
| i truly enjoyed all the action and the characters in this book the interactions between all the characters keep you drawn in to the book | 5/5 | 0.00066 |
| this book was the worst zombie book ever not even worth the review | 1/5 | 1.00627 |
| way too much sex and i am not a prude i did not finish and then deleted the book | 1/5 | 1.00635 |
| in reality it rates no stars it had a political agenda in my mind it was a waste my money | 1/5 | 1.00742 |
| fortunately this book did not cost much in time or money it was very poorly written an ok idea poorly executed and poorly developed | 1/5 | 1.00812 |

∼6k turns in each set (training and test) for task 1 and ∼9.5k turns in each set for task 2. Here, we consider the training and test data associated to tasks 1 and 2 because the other tasks require querying Knowledge Base (KB) upon user request: this is out of the scope of the paper.

In task 1, the user requests to make a new reservation in a restaurant by defining a query that can contain from 0 to 4 required fields (cuisine type, location, number of people and price range) and the bot asks questions for filling the missing fields. In task 2, the user requests to update a reservation in a restaurant between 1 and 4 times.

The training set is built in such a way that, for each turn in a dialogue, together with the positive (correct) response, 100 possible negative responses are selected from the candidate set (set of all bot responses in the Facebook bAbI dialogue dataset with a total of 4212 samples). The test set is built in such a way that, for each turn in a dialogue, all possible negative responses are selected from the candidate set. More precisely, for task 1, the test set contains approximately 6k positive and 25 million negative dialogue history-reply pairs, while for task 2, it contains approximately 9k positive and 38 million negative pairs.

For our autoencoder, we use a gated recurrent unit (GRU) [4] with 1024 hidden units and a projection layer on top of it in order to replicate the input sequence in output. An upper limit of 100 was set for

the sequence length and a feature size of 50 was selected for word embeddings. The GRU uses ReLU activation and a dropout of 0.1. This model is implemented in Tensorflow and trained with the adam optimizer (learning rate of 0.0004) and a mini-batch size of 100 samples.

In this experiments, our network equals the state-of-the-art performance of memory networks presented in [2] by achieving 100% accuracy both for next response classification and for dialogue completion where dialogue is considered as completed if all responses within the dialogue are correctly chosen.

## 5    Conclusions

We have introduced a simple energy-based model, adversarial regarding data by minimizing the energy of positive data and maximizing the energy of negative data. The model is instantiated with autoencoders where the specific architecture depends on the considered task, thus providing a family of RCNs. Such an approach can address various covariate shift problems, such as not-in-training and positive and unlabeled learning and various types of data.

The efficiency of our approach has been studied with exhaustive experiments on CIFAR-10, CIFAR-100, the Amazon reviews dataset and the Facebook bAbI dialogue dataset. These experiments showed that RCN can obtain state-of-the art results for the dialogue completion task and competitive results for the general $A/\neg A$ classification problem. These outcomes suggest that the energy value provided by RCN can be used to asses the quality of response given the dialogue history. Future works will extend the RCN to the multi-class classification setting.

These results suggest that the energy value provided by RCN can be used to assess the quality of the response given the dialogue history. We plan to study further this aspect in the near future, in order to provide an alternative metric for dialogue systems evaluation.

**Acknowledgments**

This work has been funded by the European Union Horizon2020 MSCA ITN ACROSSING project (GA no. 616757). The authors would like to thank the members of the project's consortium for their valuable inputs.

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
