[Reviews · NeurIPS 2017]

Reviewer 1



The paper proposes a variant of learning for binary classification with a score that characterizes the difference between an auto-encoder's reconstruction and the original sample. The idea seems interesting and has some novelty. One concern is that binary classification is a very limited setting and there are very few binary problems in the real world. Based on the formulation, it looks like it should be easy to give a discussion how it can be applied to multi-class classification or other problems. Experiments should also be conducted. Meanwhile, the experiments are relatively weak and on relatively small datasets. The reconstruction result does not look sharp either. Considering the tremendous progress the community has made in recent years on GANs, there should be many ideas that on can borrow to produce better results. In summary, the idea in the paper looks interesting, but the discussion and experiments in the paper may not be sufficient for publication yet. I encourage the authors to further the idea a bit and submit again.

Reviewer 2



The paper proposes a classification technique using deep nets to deal with: (a) covariate shift (i.e., when the train and test data do not share the same distribution) (b) PU settings (i.e., when there are only positive and unlabeled datapoints are available). The key idea is to train an auto-encoder that reconstruct the positive instances with small error and the remaining instances (negative or unlabeled) with a large error (above a desired threshold). This structure forces the network to learn patterns that are intrinsic to the positive class (as opposed to features that are discriminative across different classes). The experiments highlight that the proposed method outperforms baselines across different tasks with different data types (image, short text, and dialogues). Overall, I enjoyed reading the paper. It is honest, concise and well-written. The idea of considering the reconstruction error of auto-encoders as an energy function is indeed interesting. While my overall recommendation is accept, there are a few points that I am not confident about: (1) The baselines; For the image classification experiments, the authors use a standard CNN as a baseline and I'm not quite certain if these count as a competitive baseline (specially for the "not-in-training" setup). My question is that are there other baselines that are more suitable for such settings? (2) The design of auto-encoders are discussed only briefly. The suggested structures are not trivial (e.g., (32)3c1s-(32)3c1s-(64)3c2s-(64)3c2-(32)3c1s-512f-1024f). The only discussion on the structure is the following sentence: "The choice of architectures for standard classifier and auto-encoder is driven by necessity of fair comparison." which is frankly not sufficient. At the same time, It seems to me this is the general trend in research on Neural Nets. Thus, I'm not confident how much this concern is valid. Overall, my recommendation is accept however with not so great confidence in my review.

Reviewer 3



This paper describes an autoencoder model which will assign low construction error for positive sample and high construction error for negative samples. For image binary classification problem, seems this autoencoder outperforms traditional standard classifier under a high inbalance data setting. For Amazon review classification problem, I notice for the given samples, positive examples are significantly shorter than negative examples, which looks like a bias. For the QA experiment, in this paper only checked 2 out of 1000 tasks and announce it achieves same performance to memory network. Overall I think this paper is ok but the technique depth and impression are not meeting the standard of NIPS.